# Matchup Characteristics of Sea Surface Salinity Using a High-Resolution Ocean Model

**Frederick M. Bingham** [1,*] , **Severine Fournier** [2] , **Susannah Brodnitz** [1], **Karly Ulfsax** [1] and **Hong Zhang** [2]

1   Center for Marine Science, University of North Carolina Wilmington, Wilmington, NC 28403, USA; brodnitzs@uncw.edu (S.B.); ku7491@uncw.edu (K.U.)
2   Jet Propulsion Laboratory, California Institute of Technology, Pasadena, CA 91109, USA; severine.fournier@jpl.nasa.gov (S.F.); hong.zhang@jpl.nasa.gov (H.Z.)
*   Correspondence: binghamf@uncw.edu; Tel.: +1-910-962-2383

**Abstract:** Sea surface salinity (SSS) satellite measurements are validated using in situ observations usually made by surfacing Argo floats. Validation statistics are computed using matched values of SSS from satellites and floats. This study explores how the matchup process is done using a high-resolution numerical ocean model, the MITgcm. One year of model output is sampled as if the Aquarius and Soil Moisture Active Passive (SMAP) satellites flew over it and Argo floats popped up into it. Statistical measures of mismatch between satellite and float are computed, RMS difference (RMSD) and bias. The bias is small, less than 0.002 in absolute value, but negative with float values being greater than satellites. RMSD is computed using an "all salinity difference" method that averages level 2 satellite observations within a given time and space window for comparison with Argo floats. RMSD values range from 0.08 to 0.18 depending on the space–time window and the satellite. This range gives an estimate of the representation error inherent in comparing single point Argo floats to area-average satellite values. The study has implications for future SSS satellite missions and the need to specify how errors are computed to gauge the total accuracy of retrieved SSS values.

**Keywords:** surface salinity; ocean modeling; representation error; satellite validation; matchups





## 1. Introduction

Since 2009, three satellite missions have been launched to measure sea surface salinity (SSS), Soil Moisture and Ocean Salinity (SMOS) from the European Space Agency, Aquarius from NASA/SAC-D and Soil Moisture Active Passive (SMAP) also from NASA. These missions utilize sun-synchronous polar orbits with high inclinations, but differing spatial and temporal resolutions, and have provided continuous SSS measurement coverage of the global ocean [1]. They all measure ocean brightness temperature at 1.4 GHz (L-band) that can be converted to SSS, a process known as retrieval [2,3]. The specifications for each of the missions is nicely summarized by Figure 2 in [4]. In terms of spatial (temporal) resolution, it is about 60 km (3 days) for SMOS, 100 km (7 days) for Aquarius and 40 km (2–3 days) for SMAP.

In order to gauge the accuracy of the satellite measurements of SSS, they are often compared to measurements taken in situ by instrumentation, a process known as validation. These may take the form of comparisons with gridded Argo products such as that of [5] or [6] ([2,3,7–12]). There may also be comparisons with individual observations such as floats, saildrones, thermosalinographs or moorings [8–10,13–20]. Comparisons may be done using satellite data at level 2 (L2) (e.g., [8,9,13]) or level 3 (L3) (e.g., [15]).

In general, in situ measurements of SSS are sparse compared to satellite measurements. For comparison at L2, in a typical year, the Aquarius satellite made 30 million L2 observations of surface salinity versus almost 100,000 observations from Argo floats, amounting to on average over 200 Aquarius L2 satellite observations for each Argo one.

One has to keep in mind here the mismatch in scale between in situ observations, which are usually made at a single point in space and time, and satellite observations, which are snapshot averages over a footprint. This introduces a representation issue inherent in the comparison of satellite and in situ observations [21–24].

The details of the validation done in the numerous studies cited above differ, and the results may depend on these details. One important detail the present study focuses on for L2 validation is how matchups are done between individual in situ observations and L2 satellite measurements for the purpose of comparison between them. That is, within the validation process, how are satellite and in situ observations matched together to form comparisons? Schanze et al. [25] discussed some ways this can be done. The one we explore here is their "ASD", or "all salinity difference", which averages all satellite observations within a space–time window to form one comparison value for each in situ observation. Other possibilities include taking the closest point in space, or time, or some weighted combination of space/time; or computing salinity difference with all available L2 observations and no averaging. Within the ASD method, the main parameters are the spatial and temporal search windows used for finding and averaging matchup values. In a couple of validation studies done using L2 satellite measurements and the ASD method [8,9,13], Argo float measurements are compared with averages of many L2 satellite values (Table 1), but there is no testing of how differences may depend on the spatial or temporal search window from which these averages are computed.

**Table 1.** Matchup criteria for three of the studies referenced.

| Reference | Comparison |
| --- | --- |
| [13] | L2 compared to Argo within 12 h and 200 km. |
| [8,9] | Searched for the closest point of approach (CPA) of the satellite to each Argo float. Time window ±3.5 days and space window 75 km; 11 L2 samples averaged for comparison with the float value. |

In this paper, we do some of the exploration started previously in [25]. Our main tool is a high-resolution (1/48°) ocean model as described below, with the assumption that the model simulates the upper ocean's spatial and temporal variability well enough to make valid conclusions about the matchup criteria we are studying; ref. [26] found that the model we use adequately simulated the ocean submesoscale as a part of the global heat budget. Beyond that, the large advantage of using a model over real data is that there is no retrieval error associated with obtaining L2 estimates from a model. That is, any differences between simulated satellite and simulated float values are the result of representation error, not errors in the corrections needed to put out actual SSS measurements as detailed by, for example, [2]. Thus, to the extent that the model does simulate the real upper ocean variability, the statistics we compute below can be considered as an estimate of representation error as well.

Thus, we will examine the space–time window for doing matchups. How large should that window be? Does time matter more than space? How sensitive are the comparison statistics, bias and RMSD (root mean square difference), to the choices made? As the space–time window decreases, and fewer L2 observations are included in the comparison, the variance of the set of L2 observations increases, and thus one might expect larger deviations from a comparison Argo float. On the other hand, as the space–time window increases, one would expect float measurements to depart further from the satellite measurements as there is a greater chance of the float finding itself within a larger scale SSS field that varies. Thus, there may be a space–time window to use for doing validation studies that minimizes the mismatch. Such a result using real in situ and satellite data was found by [25]. At the very least when considering the errors in SSS remote sensing, one should be aware of these tradeoffs and how they might impact the total error.

The issues studied here especially pertain to potential new SSS satellite missions. For new missions (or even existing ones) it is essential to not only characterize the error structure properly, but to specify how the errors are to be computed. Requirements for satellites are determined before launch (e.g., [27]). The extent to which the final mission can adhere to these requirements, as we will see in this paper, depends on the details of how the errors are computed.

## 2. Data and Methods

We make use of a high-resolution numerical model to study matchup tradeoffs. This is done by simulating Argo and L2 satellite observations. The model SSS field was taken from the evaluation time period, 1 November 2011 to 31 October 2012.

### 2.1. Observations

#### 2.1.1. L2 Aquarius

We use Aquarius L2 observations [28,29], only those points with land fraction less than 0.5%, and between 68° S and 54° N, according to the limits of the model. There were about 27 million Aquarius L2 observations during our one-year study period. We use mainly just the tracks, not the actual observations, though one week of real satellite observations is shown in Figure 1b.

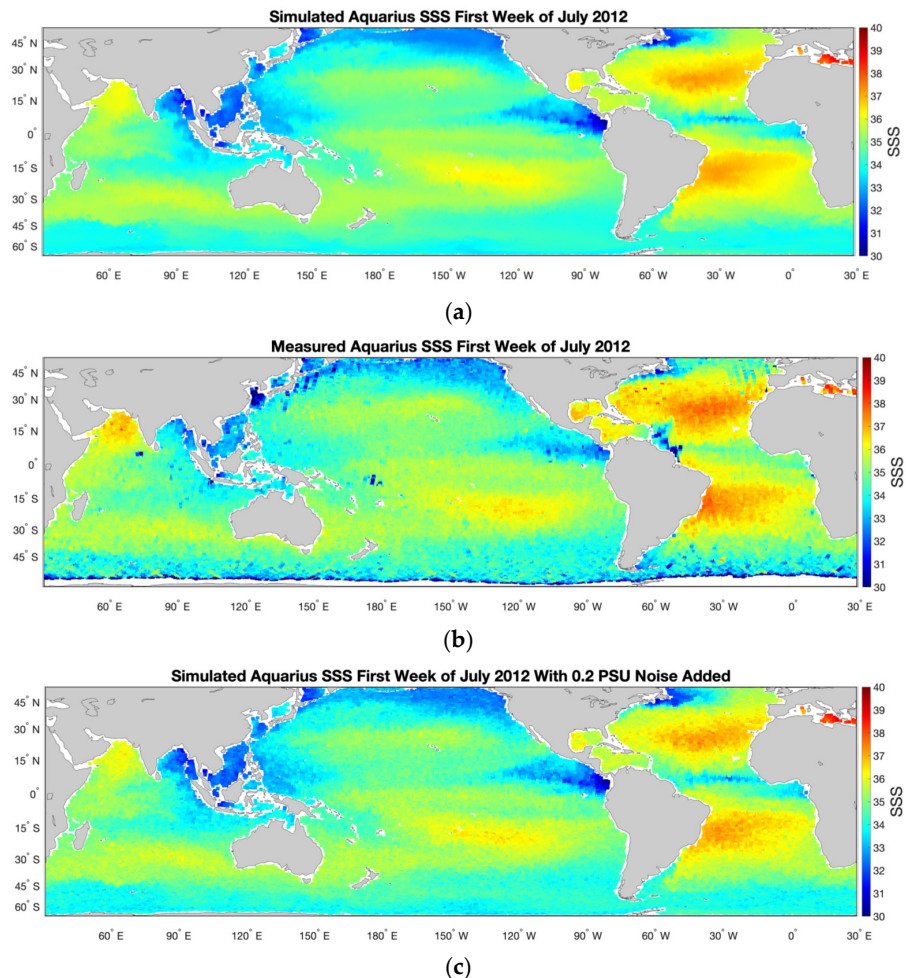

**Figure 1.** Aquarius L2 SSS observations during the first full week of July 2012. (**a**) Simulated values from the model. (**b**) Measurements from the Aquarius satellite. (**c**) Simulated values from the model with Gaussian random noise added with a standard deviation of 0.2. Color scale is at the right for each panel.

Aquarius had three parallel beams along the satellite track [28,30], sampled every 1.44 s. Figure 2a shows a single Argo float observation and a set of matching Aquarius tracks and L2 samples. A single ascending satellite pass, with three beams, is seen going southwest–northeast in the figure, with the float observation situated between two of the beams. The total coverage in 7 days, and 2 days, surrounding a float observation is included in the figure; also seen are the observations within 2 days and a 200 km radius. The green points in the figure would be averaged together to get a single ASD value to compare with the float observation at the 2 day/200 km search window. The particular float observation shown, from the North Atlantic poleward of the subtropical SSS maximum, is in a region where SSS increases to the north by about 0.2–0.4 within the 200 km sample radius (Figure 2b). The distribution of SSS within the 200 km radius has a typical long low tail and sharp cutoff on the high side (Figure 2c; [31]). The ASD mean value of SSS from those L2 observations (black line in Figure 2c) is lower than the float value (red line) by about 0.1. The same computation can be done with all the available simulated Argo floats and combined to get an RMS difference, as will be displayed below.

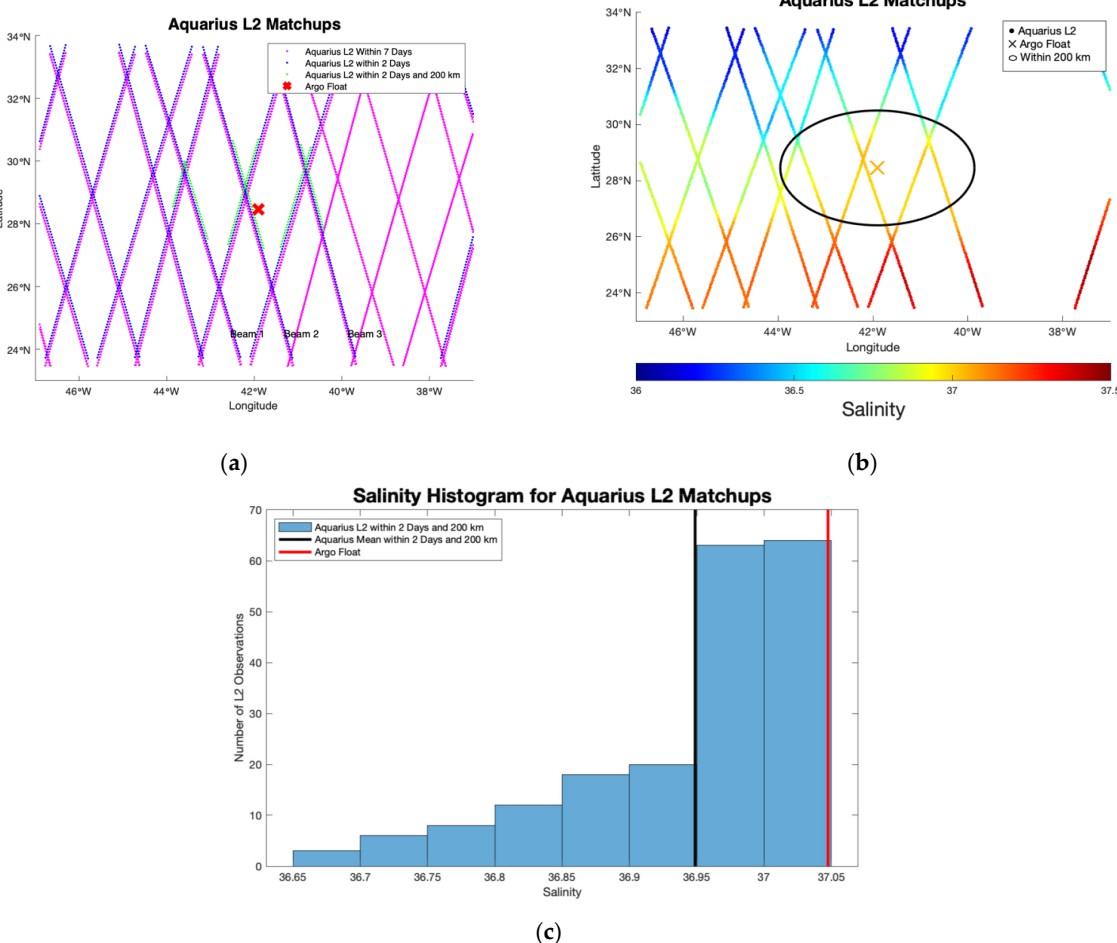

**Figure 2.** (**a**) Red "X": Argo float that surfaced at the indicated point on 28 November 2011 at 10:10:33. Magenta symbols: Aquarius L2 observations within 7 days and 5° of the float. Blue symbols: Aquarius L2 observations within 2 days and 5° of the float. Green symbols: Aquarius L2 observations within 200 km and 2 days of the float. Beams are indicated with black text. Blue and green symbols are shifted slightly for clarity. (**b**) "X": The same Argo float colored by model salinity with the scale at the bottom of the plot. Small circular symbols: Aquarius L2 observations within 2 days and 5° of the float, colored by salinity with the scale at the bottom of the plot. Black ellipse shows the area within 200 km of the float. (**c**) Histogram of simulated Aquarius L2 salinity within 2 days and 200 km of the Argo float from panel (**a**). The mean of the simulated Aquarius L2 values is shown in black and the simulated float SSS value is shown in red.

### 2.1.2. L2 SMAP

We also extracted the L2 observation points from the SMAP data [32]. SMAP was not launched until 2015. Since our evaluation time period was 2011–2012, which is the period model output was available, we simply subtracted 4 years from the time of each SMAP L2 observation to match the time span of the model. We do not use the actual satellite SMAP L2 observations in this study, only their locations and times. Thus, the SMAP tracks we used were from the 1 November 2015–31 October 2016 time frame. There were about 142 million observations. The SMAP L2 dataset is much larger than one from Aquarius because it is collected in a different way. SMAP data are averaged onto an approximately 25 km × 25 km grid that surrounds the nadir point of the satellite track (Figure 3). There are two standard versions of the SMAP data, the JPL (Jet Propulsion Laboratory) and RSS (Remote Sensing Systems), with differently structured L2 grids. We chose to use the RSS version [32], shown in Figure 3b. Similar to Aquarius, we used only observation points with land fraction < 0.5% and between 68° S and 56° N. (The limits are slightly different for SMAP than for Aquarius because of the way the L2 simulation is done.)

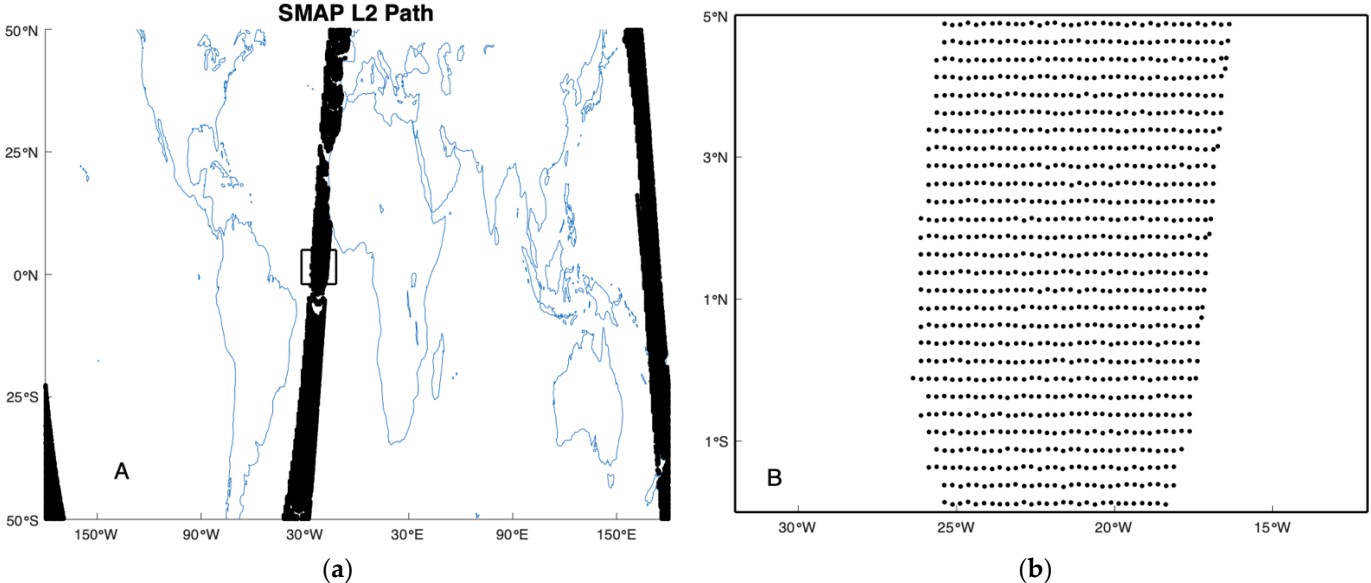

(**a**)　　(**b**)

**Figure 3.** (**a**) A sample path of the SMAP satellite showing the locations of L2 observations for one swath. (**b**) A zoomed-in view showing the L2 observation grid from a small box near the equator indicated in panel (**a**).

### 2.1.3. Argo

The Argo data we used were downloaded from the Argo data assembly center at the National Centers for Environmental Information for the evaluation time period. We used only the locations and times of the float surfacings, not the measurements themselves. However, to make certain we had a good dataset, we only made use of Argo data where the salinity quality flag had a value of "1", and the shallowest observation was at 10 dbars pressure or less. The Argo dataset has about 98,000 observations (Figure 4).

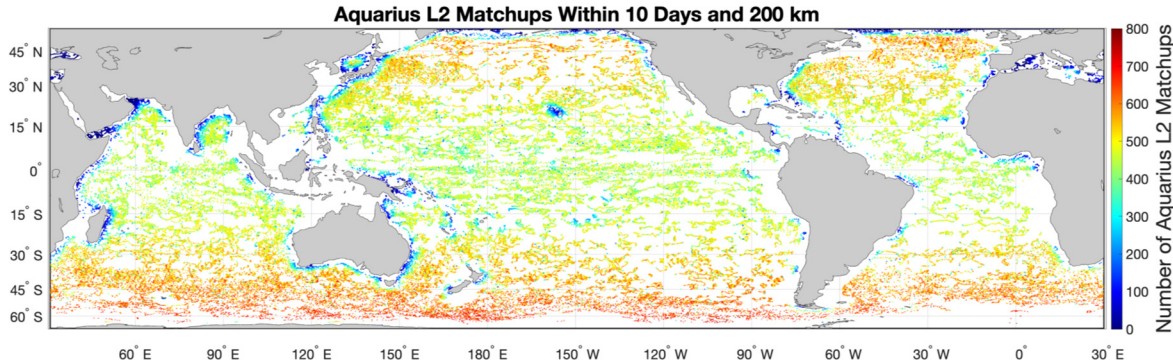

**Figure 4.** Locations of float observations used in this study. Colors indicate the number of Aquarius L2 observations within 10 days and 200 km of each float with scale at right.

### 2.2. The MITgcm

The model we use is the MITgcm, the ocean general circulation model from the Massachusetts Institute of Technology. The model is on a latitude–longitude polar cap (LLC) grid, between the latitudes of 70° S and 57° N. We use the LLC4320 version with a nominal horizontal spacing of 1/48° forced by six-hourly surface atmospheric fields from the ECMWF (European Centre for Medium-Range Weather Forecasting) operational atmospheric analysis [26]. There are about 14 months of model output available, but we use exactly one year, the evaluation time period, 1 November 2011 to 31 October 2012. Our analysis is with hourly output, though the model has a shorter time step than that. The model is free-running; it does not assimilate any ocean data.

### 2.3. Simulation Data

Simulated satellite and Argo data were generated by sampling the model at real world observation locations and times. This section describes in detail how that process was executed.

### 2.3.1. Simulated Satellite L2

The model was sampled as if the satellite were flying over it. We took the L2 tracks from the Aquarius and SMAP satellites (e.g., Figures 2a and 3) and superimposed them on the model. For Aquarius, the footprint is about 100 km in diameter [30], and each L2 observation is a weighted average with half-power point at 50 km radius from the center of the footprint. SMAP SSS L2 values are similar, with a 20 km footprint radius [2].

At every point in time and space where there was a satellite L2 observation of SSS, we created a simulated one at the closest hourly model time step. This was done by looking at a 100 km (40 km for SMAP) distance surrounding each L2 observation, the light and dark blue areas in Figure 5. We took all of the model grid points within that area (the "evaluation region") and computed a Gaussian-weighted average of those points,

$$S_{L2} = \frac{\sum_C w_i S_i}{\sum_C w_i} \tag{1}$$

where $S_i$ is the set of gridded model SSS values (located at the red dots in Figure 5) within 100 km (40 km for SMAP) of the L2 observation point. $S_{L2}$ is the simulated L2 value at the same point (yellow dot). The summation, C, is done over the set of model grid points within the evaluation region, i.e., the red dots within the light and dark blue areas in Figure 5. The $w_i$ are weights given by

$$w_i = e^{-\ln(2)\left(\frac{d_i}{d_0}\right)^2}, \tag{2}$$

where $d_i$ is the distance between the L2 observation point and each model grid node and $d_0$ is 50 km (20 km for SMAP), the footprint radius. The weighting function is such that its value is 0.5 at a distance equal to the footprint radius.

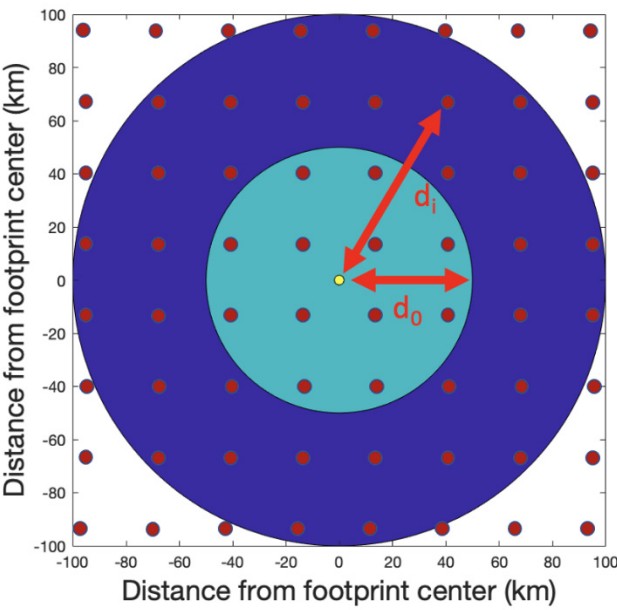

**Figure 5.** Diagram showing how the L2 simulation is done for the Aquarius satellite. The yellow dot in the center is the location of an L2 observation. The red circles are example model grid nodes—the ones from the actual model are much more densely packed. The footprint radius, $d_0$, is 50 km, and $d_i$ is the distance from the evaluation point to a sample model grid node used in the computation of $w_i$ in Equation (2). The light blue region is the 100 km diameter footprint. The dark blue region contains more model grid nodes used in the computation of the simulated value. The summation, C, in Equation (1) is over all model grid nodes within the light and dark blue regions.

Figure 1a,b compares one week of model output with one week of real Aquarius observations. One can see there are differences in the details—there is no expectation that these would match exactly as no ocean data are assimilated by the model, and the purpose of this exercise is to use the idealized environment of the model, free of retrieval errors, to test matchup parameters. The simulated fields are smoother and less noisy, especially at high latitudes. However, the basic features of the real data are reflected in the simulated data, the high SSS subtropical regions in each ocean basin, the eastern Pacific fresh pool, the contrast in SSS between Atlantic and Pacific basins, etc. The high SSS subtropical regions in all the ocean basins are saltier in the measured values than the model, and fresh regions such as the eastern Pacific fresh pool, western Pacific, South China Sea and Bay of Bengal. A large difference between the panels is that the low SSS signature of the Amazon outflow in the real observations is almost absent in the model. This is likely because the model uses monthly climatological runoff values from [33] for river input [34], which may be very different from the actual discharge of the Amazon in July 2012.

### 2.3.2. Simulated Argo

In addition to the simulated satellite data, we put together a simulated Argo dataset. We took all the Argo data described above and sampled the model as if the float had popped up to the surface at its designated time and location. That is, we sampled the model at the closest grid node and hourly value to that of each float.

### *2.4. Matchups*

To study matchup criteria, we used an ASD approach as described above [25]. That is, we matched each individual float measurement with a set of L2 observations. The L2

observations were taken within a given time and space window and averaged to form one value for comparison. For example, Figure 2a shows the set of matchup satellite observations for one particular float at a distance of 200 km and a time of 2 days—the green symbols and observations within the oval in Figure 2b. Figure 6 shows a similar set of matchup values for SMAP. The time window indicates a total before and after difference. That is, a time window of 5 days indicates all data from 5 days before the float observation to 5 days after.

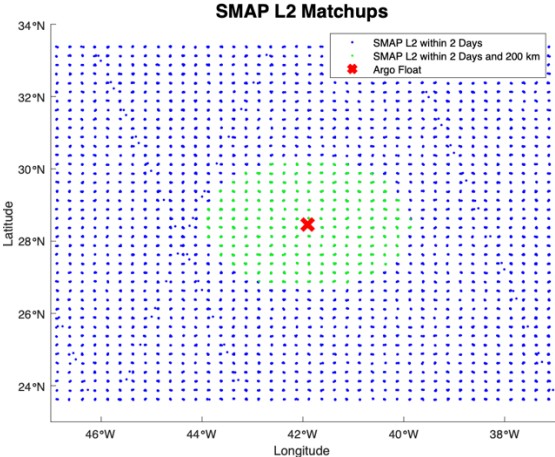

**Figure 6.** Red "X": Argo float depicted in Figure 2. Blue symbols: SMAP L2 observations within 2 days and 5° of the float. Green symbols: SMAP L2 observations within 200 km and 2 days of the float.

Figure 4 shows the number of matchups for each Argo float for the broadest criterion (10 days and 200 km). This is the number of L2 observations averaged together to form a comparison value for each float. The number mainly depends on latitude, and is smaller at low latitudes. Near coastlines and islands, the number is also lower. This is due to the land fraction criterion used to filter the L2 observations—the closer one is to land, the fewer valid L2 observations there are. At high latitudes there are more matchups per float. This is because the satellite tracks tend to get closer together and denser with increasing latitude.

The averages (the agglomerations of L2 values) were used to compute statistical measures of offset: RMSD (root mean square difference) and bias (mean difference). That is, the RMSD is the RMS of the differences between the ~98,000 individual Argo measurements and the matched set of averaged L2 satellite measurements.

The median number of satellite L2 observations averaged together per float as a function of search radius and time window is shown in Figure 7a,b. As expected, the number increases with both distance and time from about 600 (2400) for Aquarius (SMAP) to near zero at short time and space windows. Comparing the color scales for Figure 7a,b, it is seen that the number of L2 observations per float for SMAP is 4–5 times that for Aquarius. This is a result of the different sampling strategies of the two satellites shown by comparing Figures 2 and 6. The rotating antenna and trochoidal sampling pattern of SMAP yields more samples than the fixed antenna swath of Aquarius [4].

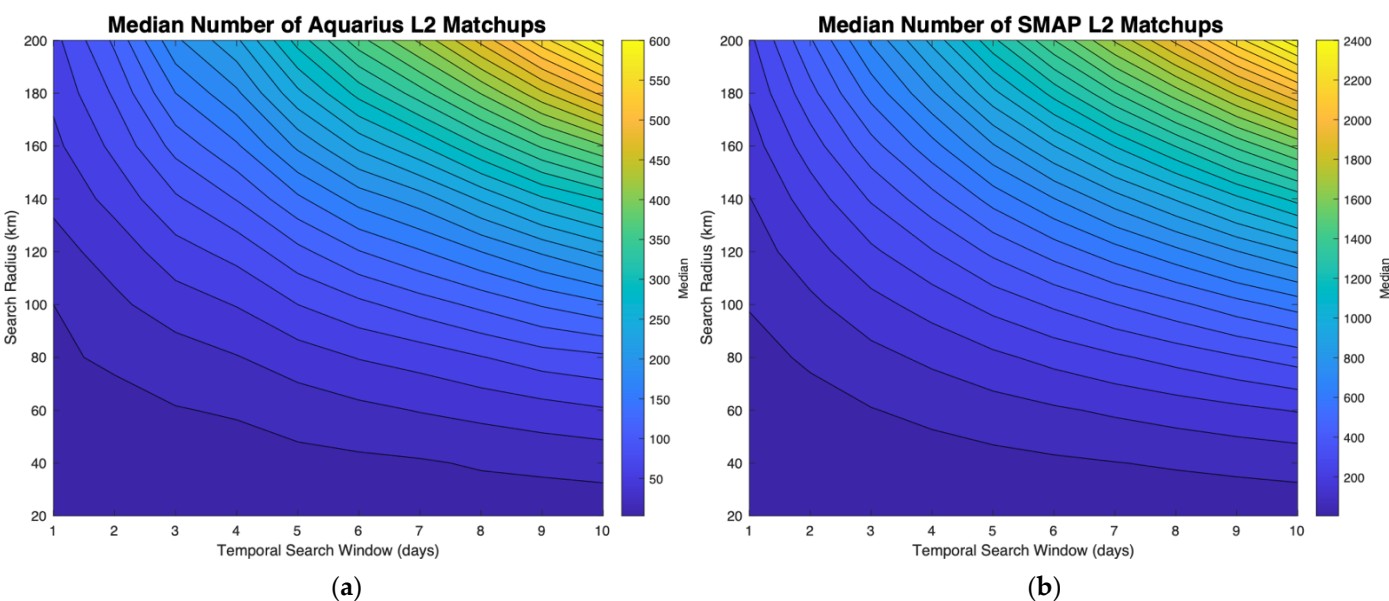

(a)             (b)

**Figure 7.** (**a**) Median number of Aquarius L2 observations per Argo float for a given temporal search window (*x*-axis) and search radius (*y*-axis). Contour lines are drawn at intervals of 20, with the lowest having a value of 20. (**b**) Same for SMAP but with contour lines drawn at intervals of 75, with the lowest having a value of 75. Note different color scales for each panel.

## 3. Results

The RMSD between simulated float and Aquarius (Figure 8a,b) shows the tradeoff between time and space search windows. For search radius less than 60 km, RMSD increases continuously as a function of time window (blue and red curves in Figure 8b). However, for search radius greater than 60 km, there is a small decrease in RMSD as a function of time window until the RMSD reaches a minimum somewhere around 3 days (yellow, purple and green curves in Figure 8b). The RMSD then increases again. In other words, for a search radius greater than 60 km, the contour lines in the figure slope upward for a short time window, and then downward for a time window longer than 3 days. This characteristic becomes clearer with increasing search radius.

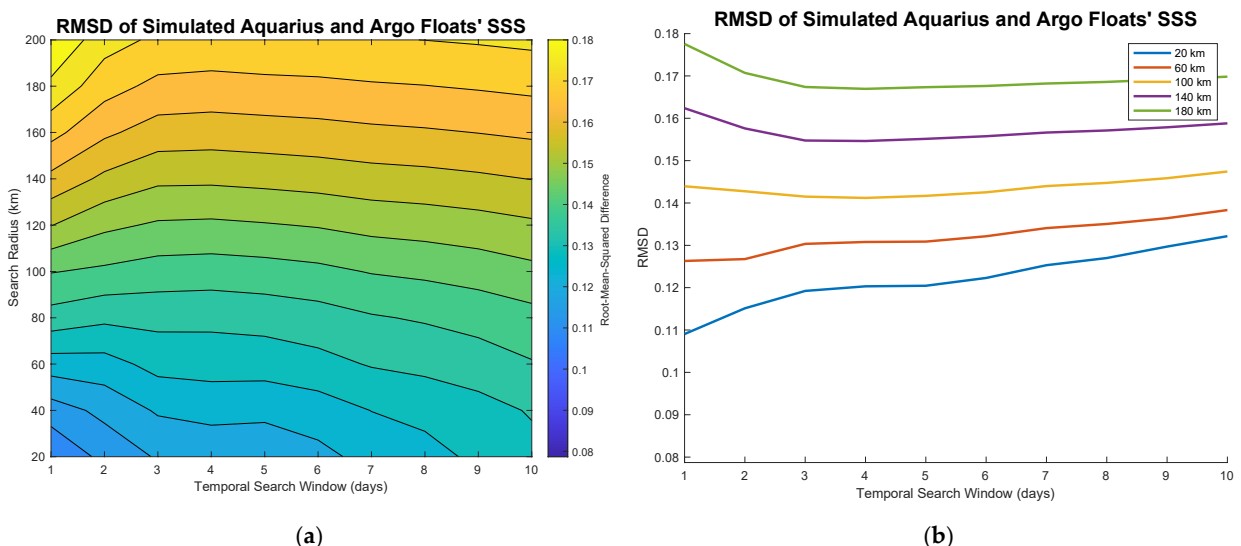

(a)             (b)

**Figure 8.** (**a**) RMS difference between simulated Argo float and average of simulated Aquarius L2 observations for a given temporal search window (*x*-axis) and search radius (*y*-axis). (**b**) The same RMS difference plotted as a function of temporal search window for 5 search radii.

This minimum RMSD at about three-day time window is the result of a tradeoff. With a short time window there are fewer measurements to average together to make reliable estimates. For a long time window there is time variability in the SSS field that generates differences between in situ and satellite values [25]. The three-day time frame appears to be just close enough to a snapshot with enough satellite measurements to make a reliable average. There is no such ideal window in space. At every time window, RMSD increases as a function of search radius; i.e., the search radius with the smallest RMSD is the one that is as small as possible. This analysis shows how the value of RMSD between satellite L2 and in situ data can vary depending on the search window chosen.

The RMSD for SMAP (Figure 9a,b) shows similar values as for Aquarius—the color scales are the same for both figures—though the RMSD values at short spatial window are smaller for SMAP. The main difference for SMAP is that the minimum RMSD is at 2 days instead of 3 (yellow, purple and green curves in Figure 9b), and is not as strong a minimum as for Aquarius. One would guess that the shorter time window for this minimum for SMAP is simply a result of having more data to produce reliable averages. As shown above, the SMAP ASD averages are made from four times more snapshot values.

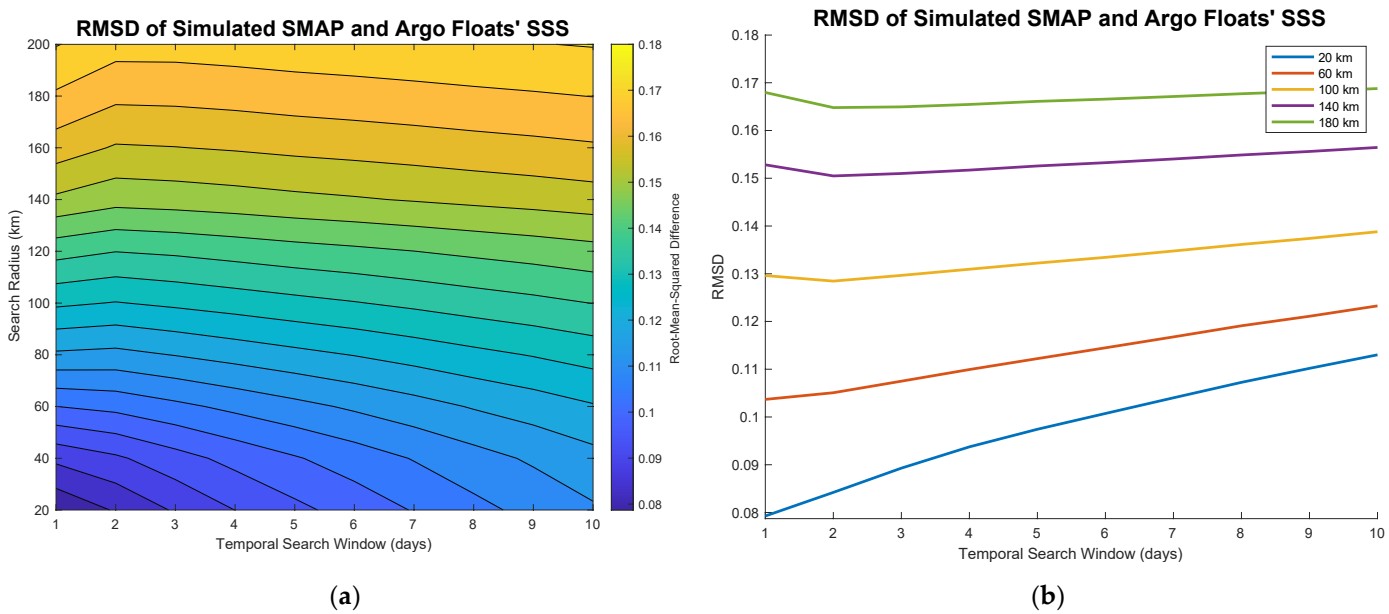

(**a**)                                    (**b**)

**Figure 9.** (**a**) Same as in Figure 8a, but for SMAP. (**b**) Same as in Figure 8b, but for SMAP.

We computed bias for all the observations as a function of space and time window. The numbers were consistently negative, meaning floats tend to take on higher values than the satellites on average. Though negative, the bias was very small, generally less than 0.002 in absolute value. We omit these plots for brevity. The fact that float SSS tends to be greater than satellite values is a consequence of the common negatively skewed distribution of SSS (e.g., Figure 2c; [31]) as explained at length by [24].

As another approach, we computed the RMSD in a different way. Given the short spatial scales of SSS [35], we computed the ASD comparison agglomerated mean L2 satellite value using a weighted average instead of a simple one. The weighting is given by a Gaussian dropoff with distance from the comparison Argo float, 50 km for Aquarius and 20 km for SMAP, the same function shown in Equation (2). The RMSD for Aquarius (Figure 10a) shows a different pattern than the non-weighted one for a larger spatial window. At distances beyond about 100 km, the RMSD shows little dependence on spatial window size. For a window less than 100 km, the RMSD is very similar to that computed with no weighting (Figure 8a). For SMAP, the results are a little different (Figure 10b). For spatial window size larger than 40 km, the weighted RMSD is much smaller than the non-weighted (Figure 9a,b) and dependence on spatial window size almost disappears. The weighting makes observations farther than

twice the footprint radius from the evaluation point essentially irrelevant. The decreased value of RMSD in Figure 10b relative to Figure 10a is likely a result of the larger number of L2 observations going into each ASD average value.

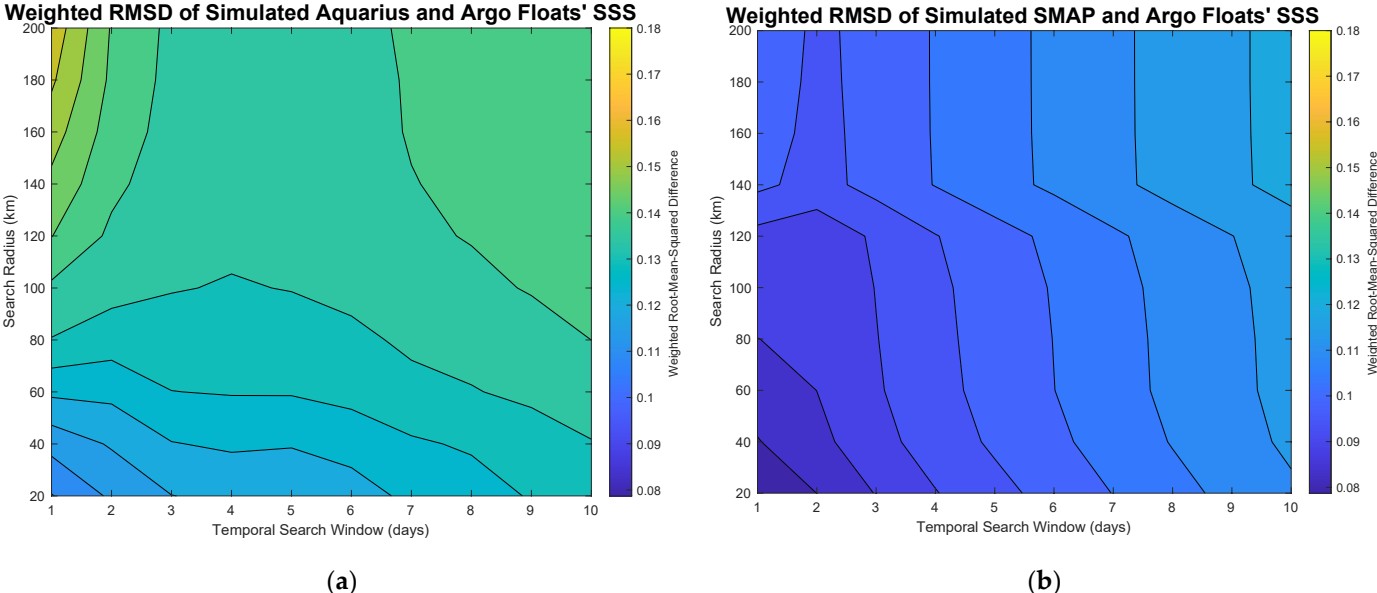

**Figure 10.** (**a**) Same as in Figure 8a, but using Gaussian weighting as described in the text. (**b**) The same as panel (**a**), but for SMAP.

The simulated L2 satellite values we used come with no "retrieval error". That is, the real satellite observation includes all the errors associated with converting raw brightness temperatures into a value of SSS, roughness corrections, galactic noise etc. [2,3,30]. The simulated values do not contain any of that, only representation error as discussed above. For that reason, we wanted to see what impact adding noise to the input data would do to the computed RMSD. Three experiments were carried out, one with normally distributed noise with zero mean and standard deviation of 0.1, another with standard deviation of 0.2 (Figure 11) and a third with 0.5 (Figure 12). That is, in computing the matchup RMSD, we first added noise to all of the L2 values used to formulate the ASD average. The simulated Aquarius SSS field for the first week of July 2012 with 0.2 noise added (Figure 1c) appears more like that of the real Aquarius data (Figure 1b) than does the no-noise version (Figure 1a). The values of 0.1, 0.2 and 0.5 are similar to those reported in some different sources. For example, [8] give a standard deviation of 0.3 for the difference between Argo and Aquarius L2 values; [36] give an error value of 0.5 for the 70 km L2 SMAP product we use here, and [12] gives values of standard deviation of 0.16 on a $1° \times 1°$ spatial scale for Aquarius.

Comparing Figure 11a,c with Figure 8a, we see that the added noise introduces a minimum in RMSD at 40 km for 0.1 noise and 60 km for 0.2 noise. For a spatial window larger than 100 km, the addition of noise makes little difference. This again highlights the tradeoff between number of observations and variability of the underlying SSS field. The ideal spatial window for the tradeoff depends on the amount of noise inherent in the field. The results for SMAP (Figure 11b,d) show no minimum in RMSD for 0.1 noise, and a minimum at 40 km for 0.2 noise. With 0.5 noise (Figure 12), the spatial minimum moves farther out to 60 km.

With noise added, the size of the time window becomes more of an issue. Rather than having almost no time dependence (Figure 8) for short space windows, the RMSD reaches a minimum at 6 days for Aquarius, vs. 2–3 days for no noise, and 3 days for SMAP (Figure 11d) for 0.2 noise, vs. no minimum at all for no noise (except at large space window). For 0.5 noise, the minimum RMSD for SMAP is at 5–6 days. The change in time window dependence seems to be a result of the number of observations to average.

**Figure 11.** (**a**) RMS difference between simulated Argo float and average of simulated Aquarius L2 observations with Gaussian noise having a standard deviation of 0.1 added for a given temporal search window (*x*-axis) and search radius (*y*-axis). (**b**) Same for SMAP. (**c**) Same as panel (**a**), but for 0.2 noise. (**d**) Same as panel (**b**), but for 0.2 noise.

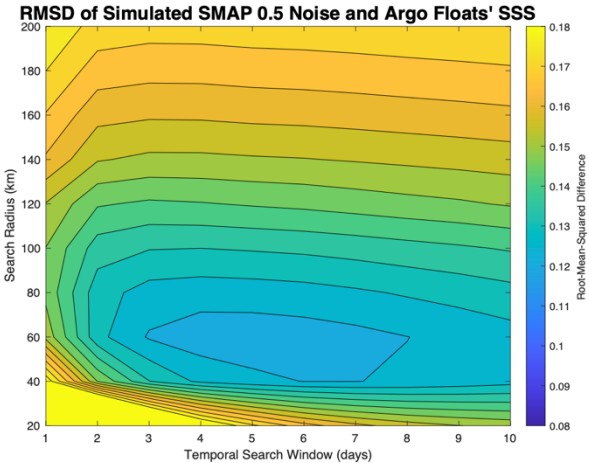

**Figure 12.** Same as in Figure 11b, but with 0.5 noise.

## 4. Discussion

In this work we looked at the way SSS is sampled by remote sensing and how that sampling is validated by comparison with in situ data at L2. We focused on the Aquarius and SMAP missions in generating the L2 comparison values from the model. These missions provide a relatively straightforward sampling pattern and footprint size. Further work in this area will focus on SMOS using a similar set of methods. The SMOS mission has a variable footprint size that depends on the look angle from nadir [4] and presents more of a challenge to simulating the L2 values. There is also the question of whether the ASD window should vary in size depending on the footprint. We speculate that the tradeoffs in space and time for SMOS will be similar to what is presented here, but do not know the magnitudes.

Schanze et al. [25] do a very similar calculation to ours, with a final recommendation to use the same "all salinity difference" method we used. Their recommendation to use a 50 km search window matches closely with the minimum RMSD seen in Figure 11. As SMAP is so much more heavily sampled and has a smaller footprint, a smaller search window than for Aquarius would be appropriate. As for the time window, [25] recommend using ±3.5 days, i.e., 3.5 days as done here. This recommendation seems about right for SMAP given the results of Figure 11, though the window might be relaxed a little to 5–7 days for Aquarius.

Keep in mind that there are many ways of validating SSS remote sensing data besides the simple one used here, as was discussed at length by [25]. One could validate at L3 instead of L2, or compare L3 values with gridded in situ products, which have their own issues of representation error [12,37], instead of individual measurements; or do the matchups a different way than the simple block averages we computed here, e.g., Figure 10. All of this is to say there is no perfect way of doing validation, only many possibilities, each with its own set of tradeoffs.

This work was carried out entirely with model data as an exercise in understanding the choices involved in doing matchups using in situ point measurements and L2 satellite values combined in a particular way. The RMSD values shown in Figure 8 and Figure 9 should not be considered any kind of overall error value for the satellite measurements. They are associated only with representation error, i.e., temporal aliasing and subfootprint variability (SFV). They do not contain any of the main sources of error inherent in satellite measurement of SSS [2]. The numbers indicated in those figures, 0.08–0.18, could be considered estimates of representation error, assuming the variability in the model is similar to that of the real surface ocean [26]. References [24,38] computed very similar numbers for representation error from in situ data at two sites in the subtropical North Atlantic and eastern tropical North Pacific at 100 km scale. Drushka et al. [23] made a global map of variability at 100 km scale from in situ data (their Figure 9a) with numbers a little larger than ours. Their values of variability at <100 km scales are about 0.1–0.25 in mid-ocean, but larger in certain areas such as western boundaries.

The main lesson to be taken from our work is that the error involved in satellite measurement of SSS depends strongly on how the evaluation is carried out. If minimizing the error is an important goal, then the weighted averaging technique demonstrated in Figure 10 is one way to do it.

One related issue we have not considered in this work is that of vertical variability of salinity. Salinity can vary quite strongly on short vertical scales, i.e., 1 m or less [22]. This can lead to a representation error similar to what we discussed in this paper, where the satellite samples the skin surface, but validation measurements sample at depth. There may be a significant difference between these for about 13% of Argo validation measurements according to [39], leading to a global average bias of −0.03. This is smaller than the RMS differences reported in Figure 11, for example, but much larger than the computed bias we reported, but for which we did not provide plots. We conclude that horizontal representation error is a larger issue than vertical salinity stratification at any space or time window when considering the satellite error budget.

One item we explored here is the tradeoff between time and space window in formulating the comparison L2 averages. The clear conclusion is that for the representation error there is only a small dependence on time in the range we examined but a strong dependence on space (e.g., Figure 8). This is a result of the unique nature of the SSS field. It has a short spatial decorrelation scale due to the influence of rainfall and submesoscale variability [23], but is dominated in time scale by the seasonal cycle in many areas [35,40,41]. This can give those doing validation studies ways to formulate their space/time search criteria for optimum results (i.e., minimum error).

**Author Contributions:** Conceptualization, F.M.B. and S.F.; methodology, F.M.B.; software, S.B. and K.U.; resources, F.M.B. and S.F.; data curation, H.Z., S.B. and K.U.; writing—original draft preparation, F.M.B.; writing—review and editing, S.F.; visualization, S.B. and K.U.; supervision, S.F.; project administration, F.M.B. and S.F.; funding acquisition, F.M.B. and S.F. All authors have read and agreed to the published version of the manuscript.

**Funding:** The work of F.M.B., K.U. and S.B. was funded by the National Aeronautics and Space Administration, grant number 80NSSC18K1322. Part of the research described in this paper was carried out at the Jet Propulsion Laboratory, California Institute of Technology, under a contract with NASA. This research was supported by NASA grant (19-OSST19-0007).

**Data Availability Statement:** Data used in this study can be obtained from the following sources: Aquarius L2. DOI:10.5067/AQR50-2IOCS; SMAP L2. DOI:10.5067/SMP40-2SOCS; Argo profiles. https://www.nodc.noaa.gov/argo/index.htm (accessed on 25 September 2020); MITgcm SSS. https://catalog.pangeo.io/browse/master/ocean/LLC4320/LLC4320_SSS/ (accessed on 1 June 2020).

**Acknowledgments:** The comments of three anonymous reviewers greatly improved the manuscript.

**Conflicts of Interest:** The authors declare no conflict of interest. The funders had no role in the design of the study; in the collection, analyses, or interpretation of data; in the writing of the manuscript, or in the decision to publish the results.

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
