# Peer review of "Matchup Characteristics of Sea Surface Salinity Using a High-Resolution Ocean Model"

_remotesensing, doi:10.3390/rs13152995_

Round 1
Reviewer 1 Report
Please see the attached comments.

Reviewer 2 Report
This paper cleverly exploits a high resolution model to quantify the representation error or matchup between Aquarius or SMAP satellite L2 SSS products and Argo observations. The methodology is well mastered, the results are clearly presented and written and are important for satellite SSS validation. I recommend publication after minor revisions, taking into account my remarks below :
The horizontal differences between satellite and Argo observations are simulated with the model but nothing is said about the vertical differences, while the authors are certainly aware that the mismatch between skin layer satellite SSS and Argo observations a few meter deep induce representation error in highly stratified regions or during rainfall (e.g. Boutin et al ., 2016). The vertical levels available in the model are not described, but they potentially allow to simulate these vertical differences. Can the authors at least discuss why this representation error has not been taken into account ?
SMOS is not included in this study but a future study with this satellite is planned in the discussion, can the autors at this stage speculate if any conclusion on SMOS representation error can be deduced from their Aquarius/SMAP analysis, or has been reported in the literature ?
How do the 0.1, 0.2, 0.5 psu simulated noise compare to the uncertainty expected from satellite SSS retrieval ?
L337 Figure 11d
L353 & 357 Schanze et al. must be replaced by [25] to follow the reference convention.
Reviewer 3 Report
In this study, the authors combined output from a GCM simulation with salinity data from satellites and Argo floats to evaluate optimal space and time window size for comparison and validation. The idea is interesting and the exercise is a useful one. However, the analysis is not deep enough in my opinion. The results are mainly presented in terms of statistics without a physical explanation in most cases.
- Perhaps its not important, but it’d be good to see a plot of the mean SSS bias in the model.
- 8 and 9. - Why are the minimum RMSD values different for Aquarius (3 days) and SMAP (2 days)? Is there a physical reason for this?
- 10 a and b: it’d good to give a physical explanation if possible for the difference sensitivity to the temporal window between Aquarius and SMAP.
- The implications of the study are also not very clear to me. One thing that comes to my mind is that none of the ocean data assimilation products include satellite salinity. It’d be great if the authors can address issues such as these from the standpoint of this study, identify any bottlenecks and suggest a pathway. Doing so will make it a more meaningful contribution.
Round 2
Reviewer 3 Report
The authors have adequately responded to my comments, I have no further suggestions for them.
Author Response
Thank you for your careful reading and your comments!